# Impact of Matrix Species and Mass Spectrometry on Matrix Effects in Multi-Residue Pesticide Analysis Based on QuEChERS-LC-MS

**DOI:** 10.3390/foods12061226

**Published:** 2023-03-13

**Authors:** Shuang Zhang, Zhiyong He, Maomao Zeng, Jie Chen

**Affiliations:** 1State Key Laboratory of Food Science and Technology, Jiangnan University, Wuxi 214122, China; 2International Joint Laboratory on Food Safety, Jiangnan University, Wuxi 214122, China; 3Joint Laboratory of Omics Technologies for Special Food, Jiangnan University, Wuxi 214122, China

**Keywords:** matrix effects, multi-residue pesticide analysis, OPLS-DA, matrix species, UPLC QTOF-MS, UPLC MS/MS, MRM, IDA

## Abstract

With the popularity of multi-residue pesticide analysis based on quick, easy, cheap, effective, rugged, and safe (QuEChERS) cleanup and liquid chromatography–mass spectrometry (LC-MS), matching optimal matrix-matched calibration protocols and LC-MS conditions to reduce matrix effects (MEs) has become a crucial task for analysts in their routines. However, dozens to hundreds of pesticide analytes in a single run generate increasingly multi-dimensional ME data, requiring appropriate tools to handle these data sets. Therefore, we established an ME analysis strategy by drawing on analytical thinking and tools from metabolomics analysis. Using this, matrix species-induced and mass spectrometry-induced systematic ME variations were distinguished, and pesticides contributed to the variations were scanned out. A simultaneous weakening of MEs on 24 pesticides in 32 different matrices was achieved using the time-of-flight-mass spectrometry (TOF-MS) scan under the information-dependent acquisition (IDA) mode of high-resolution mass spectrometry (HR-MS), compared to multiple reaction monitoring (MRM) scanning by tandem mass spectrometry (MS/MS). Bay leaf, ginger, rosemary, *Amomum tsao-ko*, Sichuan pepper, cilantro, *Houttuynia cordata*, and garlic sprout showed enhanced signal suppression in the MRM scan for 105 differential MRM transitions for 42 pesticides and in IDA mode for 33 pesticides, respectively. This study revealed the interference of matrix species and mass spectrometry on MEs and provided a novel strategy for ME analysis.

## 1. Introduction

Multi-residue pesticide analysis based on liquid chromatography–mass spectrometry (LC-MS) with quick, easy, cheap, effective, rugged, and safe (QuEChERS) cleanup plays a crucial role in global food safety regulations as the preferred method for routine laboratories. However, matrix effects (MEs) are a significant drawback of this methodology [1,2,3,4]. MEs are phenomena where the mass spectral signal of a target at the same concentration differs between the sample injection and the solvent injection [1]. MEs occur frequently and affect important method parameters, including the limit of detection (LOD), the limit of quantification (LOQ), linearity, accuracy, and precision [1,5,6]. Modifying the original QuEChERS protocol, chromatography, or spectrometry can reduce the strength of MEs but cannot eliminate MEs [3,4,5,7,8,9,10,11,12]. ME analysis is still indispensable in the analysis workflow.

Multi-residue pesticide analysis deals with tens to hundreds of analytes in a single run, with the ME of each analyte being widely variable [3,5,13]. To our knowledge, there are two ME analysis objectives for routine laboratories. In method validation, ME analysis aims to quantify ME strength and evaluate the impact of the significant MEs on the method performance, which has been satisfied by previous strategies [1,4]. In method optimization, ME analysis aims to dig out pesticides that contribute to the variations induced by different matrix species and method conditions, which appropriate strategies and tools still need to fully meet. The compensation of MEs in routine analysis relies on the matrix-matched (MM) calibration [2,4]. To reduce the time and economic cost of the preparation and detection of the MM calibration curves, several matrix typing strategies have been proposed to evaluate or calibrate MEs during multi-pesticide residue analysis, focusing on representative matrix-matched calibration for matrices of similar properties or nature and calibration using stable isotope-labeled internal standards [14,15,16,17]. However, in regulatory monitoring programs, laboratories often deal with random combinations of samples and pesticides with different natures, due to differences in the scope of sampled commodities, seasons, geographic areas, etc. [1,2,18]. Thus, a matrix typing methodology is needed to investigate the stable and specific correlations between MEs and the matrix species. On the other hand, the impact of different analytical parameters on MEs induced by a given matrix species has been widely reported. Still, the impact on MEs induced by different matrix species has yet to be studied, due to the lack of appropriate strategies [4,5,6,7,8,9,10,12,13,19,20,21,22,23].

Metabolomics analysis, an analytical profiling technique for measuring and comparing large numbers of metabolites in biological samples, provides a new perspective for ME analysis [24]. Metabolomics analysis reveals the changes in metabolites produced by an organism under different conditions, and metabolic phenotypes (metabotypes) are characterized by the profiling of metabolites [24,25]. ME analysis studies the variations of MEs on dozens or hundreds of pesticide targets induced by different conditions. Would MEs characterize ME types? Inspired by this thought, we tried to establish a ME analysis strategy base on the ideas and tools of metabolomics analysis.

In global food regulation and monitoring, pesticide analysis is mainly applied to commodities of plant origin [11,18,25,26,27,28]. In Document No SANTE/11312/2021, commodities of plant origin were grouped according to their water, acid, sugar, oil, starch, protein, and fat content, and typical representative commodities for each category were also listed [2]. The GB 23200.121-2021 National Food Safety Standard categorizes commodities as vegetables, fruits, cereal/oil seeds, condiments, edible fungi, nuts, beverages, or medicinal plants [4]. Therefore, 32 commodities of plant origin were selected as typical reprehensive matrix species according to 3 classifications, as shown in Appendix A. According to botanical classification, these matrices were from 18 families. According to Document No SANTE/11312/2021.26 and the National Food Safety Standard, matrices of the same category produced aggregation on the score plots, which were divided into 8 and 5 groups, respectively [2,4]. Their matrix species-induced ME variations were analyzed in this study.

Tandem mass spectrometry (MS/MS) in multiple reaction monitoring (MRM) scan mode, coupled with ultra-high performance liquid chromatography (UHPLC), is the most developed technique for routine pesticide analysis [1,2,4]. Quadrupole time-of-flight mass spectrometry (QTOF-MS) in information-dependent acquisition (IDA) mode coupled to UHPLC has been applied in wide-scope screening and the quantification of pesticides in food samples [12,15,29,30,31]. IDA mode consists of two experiments, the TOF-MS survey scan and the MS/MS scan. The two scans are automatically switched during acquisition to obtain the MS/MS spectra of the most intense precursor ions in a run. The pesticide targets are identified using retention time (*t*_R_), accurate measurement criteria, and the characteristic MS/MS fragmentation spectra and are quantified using the TOF-MS survey scan by the extracted ion chromatogram (XIC) of precursor ions with a narrow window. Therefore, the MRM scan and IDA mode were employed in this study for quantification purposes, and the mass spectrometry-induced MEs were discriminated and analyzed in this study. To minimize the influence of chromatography on MEs, both MS/MS and QTOF MS were equipped with the same brand and type of UHPLC, using identical chromatographic conditions.

This study established a novel strategy for ME analysis based on metabolomics analysis thinking and tools. The impact of different matrix species and mass spectrometry on MEs was investigated and verified using 73 pesticides and 32 different matrix species using the MRM scan and IDA mode, respectively. ME type was organized and characterized by MEs of pesticide targets using principal components analysis (PCA). Differential pesticides were scanned and validated using orthogonal partial least squares discriminant analysis (OPLS-DA).

## 2. Materials and Methods

### 2.1. Chemicals and Food Materials

Analytical standards of 73 pesticides (≥98% purity) listed in Appendix A were obtained from Alta Scientific (Tianjin, China) and J&K Chemicals (Beijing, China). All solutions were stored at −20 °C when not used. Acetonitrile (MeCN) of MS or HPLC grade was acquired from Fisher Scientific (Pittsburgh, PA, USA), and formic acid (>98% purity) was obtained from Sigma-Aldrich (Saint Louis, MO, USA). Deionized water (18.2 MΩ-cm) was prepared using a Merck Milli-Q IQ 7000 purification system (Pittsburgh, PA, USA). Bond Elut QuEChERS products were selected according to the National Food Safety Standard and purchased from Agilent Technologies for cleanup [4].

Food materials of 32 different species were purchased from a local supermarket and were certified organic foods. The category information of 32 matrix species was listed in Appendix A, according to the GB 2763-2021 National Food Safety Standard, the Document No. SANTE/11312/2021, and botanical classification, respectively [2,4].

### 2.2. Sample Preparation

As shown in Appendix A, QuEChERS cleanup for samples of the 32 matrix species was performed according to the National Food Safety Standard [4]. Seven matrices were prepared according to the QuEChERS procedure for light-colored fruits, vegetables, and mushrooms, including Chinese yam, lemon, maize, cabbage, oyster mushroom, and shiitake mushroom (light-green flow chart in Appendix A). Fifteen matrices were prepared according to the QuEChERS procedure for dark-colored fruits and vegetables, including wheatgrass, amaranth, *Artemisia selengensis*, garlic sprout, red chili, green chili, green pimiento, okra, ginger, asparagus, cowpea, wing bean, pea seedlings, blueberry, and orange (dark-green flow chart in Appendix A). Nine matrices were prepared according to the QuECHERS procedure for condiments and tea, including cilantro, basil, mint, bay leaf, Sichuan pepper, *Amomum tsao-ko*, *Houttuynia cordata*, rosemary, and green tea (orange flow chart in Appendix A). Soybean was prepared according to the QuEChERS procedure for oil seeds (blue flow chart in Appendix A).

### 2.3. Analysis

The UHPLC separation was performed with a Sciex (Framingham, MA, USA) UHPLC, fitted with an AQUITY UPLC^®^ BEH C18 column (100 × 2.1 mm, 1.7 μm) from Waters (Milford, MA, USA) at 40 °C oven temperature. The flow rate was 0.3 mL/min, and the injection volume was 2 μL. Mobile phase A was water with 0.1% formic acid, and mobile phase B was MeCN. The gradient for mobile phase B was started at 5%B; 1 min later, %B was linearly raised to 30%B for 1min; after that, it was linearly raised to 98% for 10.0 min, held for 3 min, then dropped sharply to 5%B for 0.1 min and held for 2.9 min.

For UHPLC MS/MS analysis, the UHPLC system was coupled with a Sciex QTRAP 5500 triple quadrupole mass spectrometer (Framingham, MA, USA) with electrospray ionization (ESI) at the positive ionization mode. MRM transitions for 73 pesticides were modified based on the National Food Safety Standard, and their detection parameters are listed in Appendix A [4]. For each pesticide analyte, the first MRM transition was used as a quantifier for the quantitation of the target compounds. The ion source worked with a temperature of 550 °C, ion spray voltage at 5500 V, curtain gas at 35 L/min, and ion source gas 1 and 2 at 60 L/min. Sciex Analyst software (Framingham, MA, USA) was utilized for instrumental control, and Sciex OS-Q (Framingham, MA, USA) software was used for data processing.

For UHPLC-QTOF-MS, the UHPLC system was coupled with a Sciex X500R quadrupole time-of-flight mass spectrometer (QTOF-MS, Framingham, MA, USA). The MS acquisition condition was modified according to Wang’s study and performed in an automatic IDA [31]. The IDA method consisted of two experiments. The TOF-MS survey scan was carried out in the m/z range from 100 to 1000, setting DP at 80 V, CE at 10 eV, and the accumulation time at 150 ms. The MS/MS scan parameters were: m/z range from 50 to 900, 10 most intense ions, excluded isotopes within 4 Da, ion tolerance at 50 mDa, and collision energy (CE) of 40 eV with a spread of ±20 eV. The working parameters of the ion source were the same as those used in the UHPLC MS/MS. The automated calibration device system (CDS) was set to perform an external calibration every five samples using calibrate solution. Sciex OS software (Framingham, MA, USA) was used for instrumental control and data processing. The theoretical m/z for the precursor ions of pesticides is listed in Appendix A; the XIC extraction width was set at 0.005 Da for the TOF-MS survey scan. An MS/MS library of 62 pesticides was established with the LibraryView^TM^ software (Framingham, MA, USA) by recording each pesticide’s MS/MS fragmentation spectrum. These MS/MS spectrums were used as reference spectra.

### 2.4. MEs

Reagent-only (RO) mixture standard and MM standard were prepared with MeCN at 50 μg L^−1^. ME was calculated as Equation (1):(1)$$\mathrm{ME}=\left(\frac{{\mathrm{Area}}_{\mathrm{MM}}}{{\mathrm{Area}}_{\mathrm{Ro}}}-1\right)\times 100$$
where Area_MM_ is the average integrated peak area of an analyte in the RO mixture standard and Area_RO_ is the average integrated peak area of an analyte in the MM mixture standard (n = 6) [32]. ME strength was classified into three levels: (a) weak ME occurred when |ME| ≤ 20%; (b) medium ME occurred when 20% < |ME| ≤ 50%; and (c) strong ME occurred when |ME| > 50% [13,33]. Due to the shift between the repeatability values, weak ME was negligible in quantification, and significant ME referred to medium or strong ME [27].

### 2.5. Multivariate Analysis

PCA and OPLS-DA were accomplished using the SIMCA 14.1 software, and unit variance scaling (UV scaling) was performed [34]. Hierarchical cluster analysis (HCA) was conducted using the OriginPro 2022b software [14]. Parameters were divided into initial clusters by squared Euclidean distances, and the resulting initial clusters were lined using the between-groups linkage method.

## 3. Results and Discussion

### 3.1. Overview of Matrix Effects (MEs) under Multiple Reaction Monitoring (MRM) Scans

MRM scans were commonly employed in the official procedures for pesticide residue analysis in food and feed [2,4,18]. A total of 372 MRM transitions of 73 pesticides were obtained using manual optimization. Further validation was performed using RO mixture standards, and 284 transitions presented good linear correlation coefficients (R^2^ ≥ 0.98). Although two transitions were commonly selected as the quantification and qualification ions during method developments based on MRM scans, we focused on the mechanistic study of the MEs instead of method validation, so 284 transitions were all employed as objects for further investigation.

The relative standard deviation (RSD, n = 6) of MEs values for each MRM transition in each matrix ranged from 0.8% to 9.7%, and the average values were employed for further analysis. A total of 9056 ME values were obtained from 284 MRM transitions in 32 matrix species. Variations of the MEs among matrix species or pesticide transitions were observed according to the color change by row or column in the heatmap (Figure 1A). An amount of 40.5% of these values indicated weak MEs, 41.2% indicated medium MEs, and 18.3% indicated strong MEs. A total of 59.5% of MRM transitions were affected by significant MEs, and matrix-induced signal suppression was more frequently observed than matrix-induced signal enhancement, as noted in previous studies [14,17,35,36]. These transitions could have been better candidates for quantifying involved pesticides. However, if no other options are available, the quantitative analysis of these pesticides requires special attention. Specifically, the MRM transitions for five pesticides, i.e., pyroquilon, ethion, triticonazole, hexythiazox, and phorate, suffered significant matrix-induced signal suppression in over 28 commodities, implying the reduced sensitivity of these pesticides in food samples. MRM transitions for six other pesticides showed extremely strong MEs (MEs > 100%) in certain matrix species, as shown in Appendix A. A similar phenomenon was reported and described in Uclés’ study for propargite in 10 matrices, including broccoli, orange, avocado, celery, cucumber, strawberry, green beans, leek, parsley, and potato [37].

The MEs of the same pesticide varied among different matrix species. For instance, the MS signal of fludioxonil was strongly enhanced in shiitake mushroom, blueberry, lemon, Chinese yam, green tea, basil, and soybeans (with ME values between 53% and 146%). However, it was significantly suppressed in rosemary, *Amomum tsao-ko*, Sichuan pepper, and ginger (with ME values between −32% and −72%). The distribution of MEs in each matrix species for each pesticide and their MRM transitions are colored by the strength of the MEs in Figure 1B, showing that significant MEs were observed in all commodities. In Sichuan pepper, bay leaf, *Amomum tsao-ko*, and rosemary, over 78% of the MRM transitions showed significant MEs, with the first two matrix species involving all 73 pesticide analytes.

### 3.2. Matrix Typing by MEs under the MRM Scan

From the distribution of MEs shown in Figure 1, the variability of MEs between pesticides and matrix species under MRM scan-based detection and their similarities were observed. The values of MEs for 284 transitions in 32 matrix species were placed in a “32 × 284” matrix. We used the matrix species as observations and the MEs of 284 MRM transitions as variables for matrix typing. After UV scaling, PCA modeling analysis was established. The R^2^ and Q^2^ values were respectively calculated as 0.73 and 0.63 using the two first components, demonstrating the suitability and predictability of PCA modeling. A total of 32 different matrix species were divided into three groups, as shown in Figure 2A, and the PCA score plot was colored according to three different categorical methods, as shown in Appendix A. The first group, expressed as group G1, included bay leaf, ginger, rosemary, *Amomum tsao-ko*, Sichuan pepper, cilantro, *Houttuynia cordata*, and garlic sprout, which are mainly spices. The second group, described as group G2, included amaranth, *Artemisia selengensis*, asparagus, basil, blueberry, cabbage, Chinese yam, cowpea, green tea, green chili, lemon, maize, mint, okra, oyster mushroom, pea seedling, red chili, shiitake mushroom, soybean, wheatgrass, and winged bean. The third group, including orange and zucchini, were considered Hotelling’s T2 outliers. Notably, consistent grouping results were obtained from the dendrogram of HCA (Figure 2B). Orange and zucchini, with 20.7% and 29.8% of MRM transitions suffering significant MEs, respectively, exhibited the lowest overall MEs among the 32 commodities. The PCA model labeled the two matrix species as Hotelling’s T2 outliers. In the first group, 68.1% to 87.7% of MRM transitions suffered significant MEs, showing the highest overall MEs among the 32 commodities. The strengthened signal suppression on pesticide targets in spices was also reported in most of the published methods, caused by the presence of ionic species, polar compounds, and organic molecules for the most part [38,39,40,41,42]. These results demonstrated that PCA and HCA could discriminate between the matrix species, depending on the MEs of the MRM transitions. To obtain information about differential pesticides corresponding to the range of MEs observed between groups, an OPLS-DA analysis was conducted.

Hotelling’s T2 outliers were removed according to the initial PCA results, and OPLS-DA was used to screen out differential MRM transitions between group G1 and group G2. The R^2^Y value of the OPLS-DA was 0.831, indicating the good explanatory capability of the model. The Q^2^ value was 0.810, and the difference between Q^2^ and R^2^Y was 0.021 (<0.3). In the 500 permutation tests of the OPLS-DA model, the intercept of Q^2^ was −0.349 (<0.05), indicating that the model was not overfitted (Figure 3B). The score plot divided the matrices into two groups, along with the predicted principal components (t [1]., Figure 3A). Using a VIP value > 1.0 and a *p*-value < 0.05 (*t*-test) as screening thresholds, 136 differential MRM transitions were obtained for 48 pesticides. According to the distribution of MEs for differential MRM transitions, the 101 MRM transitions for 42 pesticides showed weak suppression ME values in group G2 but showed significant signal suppression in group G1 (Figure 3C). As shown in Appendix A, 67 MRM transitions of 19 pesticides were involved. In addition, 14 MRM transitions of 5 pesticides, including edifenphos, methidathion, prothiofos, tau-fluvalinate, and butyl fluazifop, showed strong matrix enhancement effects in group G2 but showed weak to medium ME levels in group G1. Varied MEs in tomato, capsicum, brinjal, and cumin were also reported in C’s study for alachlor, chlorpyrifos, dimethoate, ethion, and quinalphos [40].

### 3.3. Overview of Matrix Effects (MEs) under Information-Dependent Acquisition (IDA) Mode

RO mixture standard solutions of 73 pesticides were analyzed, using the IDA mode, with UPLC-QTOF-MS. Among them, 62 pesticides showed a linear correlation, with signal-to-noise (S/N) >10 at a 50 μg/L addition. The other 11 pesticides were disulfoton, fenitrothion, fenobucarb, fenpropathrin, phorate, phorate sulfone, phorate-oxon sulfoxide, piperonyl butoxide, propargite, triadimenol, and triticonazole. The RSD (n = 6) of ME values for each pesticide analyte in each matrix ranged from 1.5% to 8.8%, and the average values were employed for further analysis. As shown in Figure 4, 1984 ME values were obtained from the 62 analytes in 32 matrix species. A total of 68.4% of these values showed weak MEs, 23.4% showed medium MEs, and 8.2% showed strong MEs. The incidence of significant matrix-induced signal suppression was 2.7 times higher than significant enhancement. In the same matrix species, pesticides of different natures showed varied MEs. As shown in Appendix A, 100% signal suppression indicated that the signal intensity of an analyte was decreased to an undetectable level with S/N < 3 in a given MM standard solution at a 50 μg/L addition. They were observed for fludioxonil in rosemary; methidathion in bay leaf and Sichuan pepper; fenchlorphos in bay leaf, *Amomum tsao-ko*, and ginger; monocrotophos in winged bean; methamidophos in green tea; and diethofencarb in Sichuan pepper. Meanwhile, as shown in Appendix A, extremely strong signal enhancement (MEs > 100%) was also observed, including difenoconazole in cowpea and basil; myclobutanil in lemon; carbendazim in cilantro, lemon, cowpea, basil, and green chili; and chlorsulfuron in 32 commodities. LODs and LOQs for these pesticides were also enhanced in the corresponding matrix species, due to the matrix-enhancement effects. Similar phenomena have also been reported in related studies, and the strong MEs could be eliminated by modifying the cleanup approach, spectrometry, or chromatography [6,10,13,17,43].

### 3.4. Matrix Typing in the TOF-MS Scan under IDA Mode

After UV scaling, PCA modeling analysis was established. The R^2^ and Q^2^ values of PCA modeling were respectively calculated as 0.619 and 0.459 using the two first components, demonstrating the suitability and predictability of PCA modeling. A total of 32 different matrix species were divided into two groups along the first component (Figure 5A), expressed as groups A and B, according to their ME types. The PCA was colored according to three different categorical methods, as shown in Appendix A. Group A included *Houttuynia cordata*, cabbage, garlic sprout, cilantro, rosemary, ginger, *Amomum tsao-ko*, Sichuan pepper, and bay leaf. Group B had the other 20 matrix species. Lemon was considered a Hotelling’s T2 outlier, due to the extremely strong MEs on flutriafol and myclobutan. As shown in Figure 4B, matrix species in group A induced medium to strong MEs more frequently than in group B. Except for cabbage, the MEs in the TOF-MS scan under IDA mode showed a similar trend under MRM detection, demonstrating the similar impact of matrix species on MEs under the two spectrometry.

Lemon was removed, according to the initial PCA result, and OPLS-DA was used to screen out differential pesticides contributing to ME variations between groups A and B. The good explanatory capability of OPLS-DA was indicated by the R^2^Y value of 0.829. The Q^2^ value was 0.798, and the difference between Q^2^ and R^2^Y was −0.031 (<0.3). The intercept of Q^2^ in the 500 times permutation tests was −0.283 (<0.05), suggesting that the model was not overfitted (Figure 5C). In the score plot, 32 matrices were separated into two groups alongside the predicted principal components (t [1]., Figure 5B). A total of 34 differential pesticides were obtained using a VIP value >1.0 and a *p*-value < 0.05 (*t*-test) as filtering thresholds, including 22 organophosphate pesticides, 4 chloroacetamide pesticides, 3 triazine pesticides, 1 thiocarbamate pesticide, 1 sulfoylurea pesticide, 1 carboxamide pesticide, 1 carbamate pesticide, and 1 amidoxine pesticide. As shown in Figure 5D, 31 differential pesticides suffered medium or strong matrix-induce suppression in matrices of group A but showed significantly decreased, mainly weak ME strength in the matrices of group B. Phosmet suffered medium matrix-induce suppression in group A but showed medium to strong matrix-induced enhancement in the 20 matrix species of group B. The strengthened signal suppression induced a greater LOQ in these 31 pesticides in group A. The weakened ME strength in the TOF-MS scan was also achieved in Meng’s study in tea samples for 114 out of 134 pesticides [44].

As shown in Figure 5E and Appendix A, the signal of chlorsulfuron was strongly enhanced in all of the 32 matrix species, and the enhancement effects in the matrices of group B were significantly stronger than those in group A. Precursor and fragment ions fit perfectly between the RO and MM mixture standard solutions in the high-resolution mass spectra of chlorsulfuron. Therefore, the enhanced MEs of chlorsulfuron in the TOF-MS scan under IDA mode were not induced by the co-eluent with the same precursor mass feature, and the mechanism needs further study.

### 3.5. Impact of Different Mass Spectrometry on MEs

To further investigate the mass spectrometry-induced differences in MEs, PCA and OPLS-DA modeling was conducted using the ME values of 62 pesticides in 32 matrices obtained under the MRM scan and the TOF-MS survey scan in IDA mode. The quantitative transition for each pesticide under the MRM scan was employed for multivariate analysis. The R^2^ and Q^2^ values of the PCA model were 0.659 and 0.535, respectively, using the two first components, and the R^2^Y value of OPLS-DA was 0.948, demonstrating the good correlation of both models. The Q^2^ value was 0.939, and the difference between Q^2^ and R^2^Y was 0.009 (<0.3). The intercept of Q^2^ in the 500 times permutation tests was −0.311 (<0.05), suggesting that the model was not overfitted (Figure 6C). The score plot of the PCA model (Figure 6A) provided an observation overview. In contrast, the score plot of the OPLS-DA model (Figure 6B) discriminated between MEs, depending on mass spectrometry, demonstrating the systematic difference. As shown in Figure 6D, 26 differential pesticide candidates were obtained using a VIP value > 1.0 and a *p*-value < 0.05 (*t*-test) as filtering thresholds. Although MEs of DEET differed significantly between the two mass spectrometries, the MEs were invariably at negligible levels in the quantitative analysis (Appendix A). Therefore, DEET was not considered a differential pesticide. The other 25 pesticides showed systematical differences in MEs between the MRM and TOF-MS survey scans, including 16 organophosphate pesticides, 2 triazole pesticides, 1 thiocarbamate pesticide, 1 sulfoylurea pesticide, 1 pyrimidine pesticide, 1 pyrethroid pesticide, 1 dinitroanilin pesticide, and 1 carboxamide pesticide.

Only one differential pesticide, chlorsulfuron, showed weak to medium matrix-induced suppression effects under the MRM scan but showed extremely strong enhancement effects under the TOF-MS survey scan in IDA mode. The MEs of the other 24 differential pesticides showed medium and strong ME strengths under the MRM scan but mainly weak strength under the TOF-MS survey scan in IDA mode. The addition levels (50 μg/L) for 13 differential pesticides were equal to or lower than MRLs in corresponding matrices, according to the National Food Safety Standard, including chlorpyrifos, coumaphos, dichlorvos, epoxiconazole, fenarimol, fenthion, hexythiazox, methidathion, pendimethalin, phosalone, phosmet, profenofos, and tau-fluvalinate. The minimum value of their S/N in TOF-MS scan values was 82.8, achieving the MRL in the TOF-MS scan [18]. The minimum value of their S/N in the TOF-MS scan values was 82.8. Their results in the MS/MS scan were also evaluated regarding determination and identification under IDA mode for screening purposes (Appendix A). Five pesticides achieved negative results in specific MM solutions, due to the loss of MS/MS data, including chlorpyrifos in *Amomum tsao-ko* and Sichuan pepper; dichlorvos in *Artemisia selengensis*; methidathion in orange, lemon, cabbage, wheat grass, cowpea, winged bean, *Artemisia selengensis*, asparagus, okra, ginger, garlic sprout, pea seedling, and zucchini; phosalone in *Amomum tsao-ko* and Sichuan pepper; and tau-fluvalinate in cabbage. The results demonstrated the matrix-induced false negatives in a multi-residue pesticide screening analysis. Further study should be conducted to overcome the loss of MS/MS data and the related false negatives in IDA mode, such as using data-independent acquisition (DIA) instead of IDA mode [45].

In IDA mode, the qualification of pesticides simultaneously relies on *t*_R_, the precursor, and fragment mass features [16,31,45]. The quantitative analysis relies on the integrated area of the XIC chromatogram with the precursor mass of each analyte, setting the XIC extraction width at 0.005 Da in the TOF-MS scan [31]. In MRM scans, quantitative analysis depends on the integrated area of the XIC chromatogram with its quantitative MRM transition [2,4]. The interference of co-eluents is minimized by the specific correlation between the precursor and daughter ions for the target [46,47,48]. According to the principles of the quadruple mass detector, both Q1 and Q3 scans for the MRM methods were low-resolution. The Q1 scan was performed to obtain the mass spectral signal of the parent ion for the pesticide analyte. The XIC chromatogram setting at 0.5 Da in the TOF-MS scan was employed to simulate the mass profiling and mass features obtained by the Q1 scan [35]. The ratio of integrated peak areas obtained by setting XIC extraction width at 0.5 and 0.05 Da was termed FC_0.5/0.05_. The distribution of FC_0.5/0.05_ reflected the difference in the amount of MS signals between the Q1 scan and the TOF-MS scan generated for the same pesticide. More mass signals were calculated as signals for pesticides in the Q1 scan when FC_0.5/0.05_ values > 1. The instrument and solvents may introduce interfering signals for pesticides in RO standard solutions with FC_0.5/0.05_ values >1 [48]. The interfering signals may also be derived from co-eluting components, mainly from matrices, solvents, and materials used during sample preparation for pesticides in MM standard solutions. As shown in Figure 7, the FC_0.5/0.05_ values for 62 pesticides in the RO standard solution ranged from 1.2 to 5.0, with phosmet at 56.4. Meanwhile, 99.3% of FC_0.5/0.05_ values for 62 pesticides ranged from 1.1 to 700.7 in different MM standard solutions, indicating more interfering signals. These interfering signals were introduced into the collision-induced dissociation process with pesticide signals and might impact the fragment behavior of pesticides [1,6]. On the other hand, the decreased ME levels in the TOF-MS scan might be related to its strong distinguishing capability on the interference and target signals [29,31,45].

These results indicated that changes in the mass spectrometry could induce the systematic variability of MEs in different matrix species, but the variability of each pesticide may differ. These results also demonstrated that the typing method could be ideally applied to spectrometry optimization for ME elimination.

## 4. Conclusions

In this study, the matrix typing strategy drawing on metabolomics analysis thinking and tools was achieved, providing an efficient approach for analysts to handle rapid ME analysis with a real-time combination of massive pesticide targets, different matrix species, and different LC-MS conditions. The overall characteristics of MEs for samples were summarized and typed using PCA analysis. The differential pesticides between groups were identified using OPLS-DA analysis and further validated by modified heatmap analysis and *t*-tests. For UPLC-MS/MS-based MRM detection and UPLC-HR-MS-based IDA detection, pesticides contributed to the matrix-induced and spectrometry-induced variety was scanned out, respectively. Bay leaf, ginger, rosemary, *Amomum tsao-ko*, Sichuan pepper, cilantro, *Houttuynia cordata*, and garlic sprout induced medium to strong suppression on 56% and 48% of pesticide targets under MRM scans and IDA mode, respectively. Switching mass spectrometry from the MRM scan to IDA mode simultaneously weakened MEs in 24 pesticides to negligible levels in 32 matrix species. Further study will focus on integrating the developed strategy with other related algorithms to realize computer-aided planning for multi-residue pesticide analysis.

## Figures and Tables

**Figure 1 foods-12-01226-f001:**
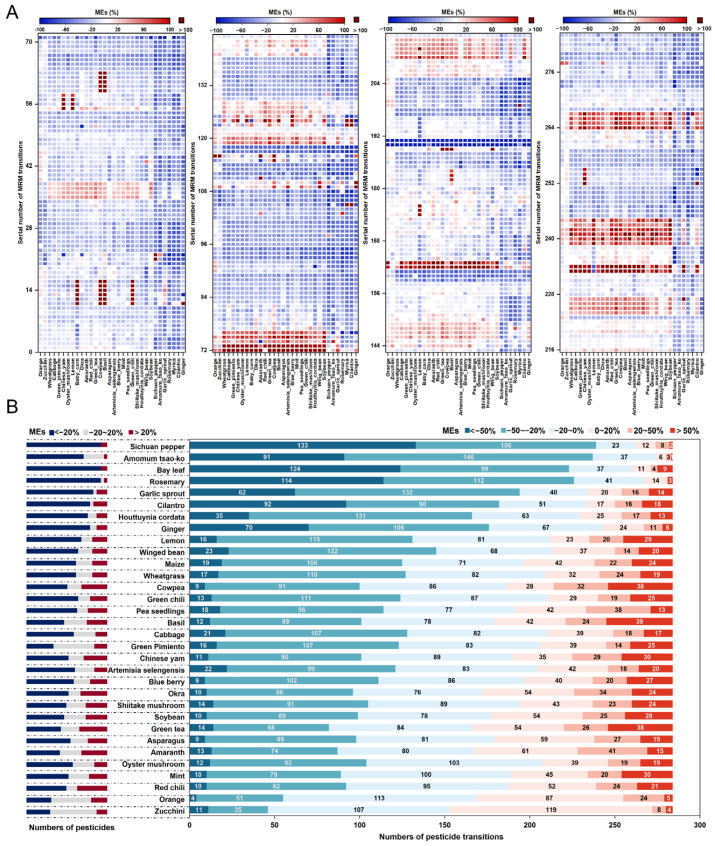
Distribution of matrix effects (MEs) on 284 transitions in 32 matrices (**A**) and the comparison of MEs in different matrix species (**B**) under the MRM scan.

**Figure 2 foods-12-01226-f002:**
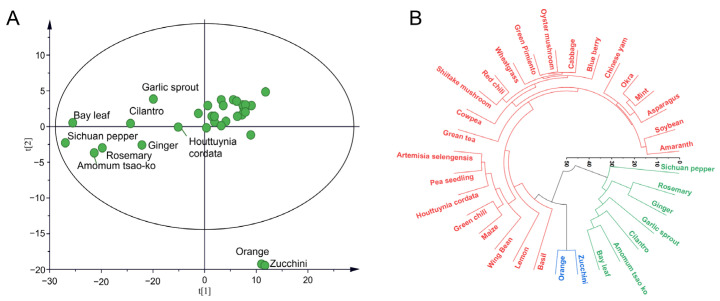
Matrix typing by MEs: (**A**) the score plots of PCA modeling; (**B**) the dendrogram of HCA colored by classifying results.

**Figure 3 foods-12-01226-f003:**
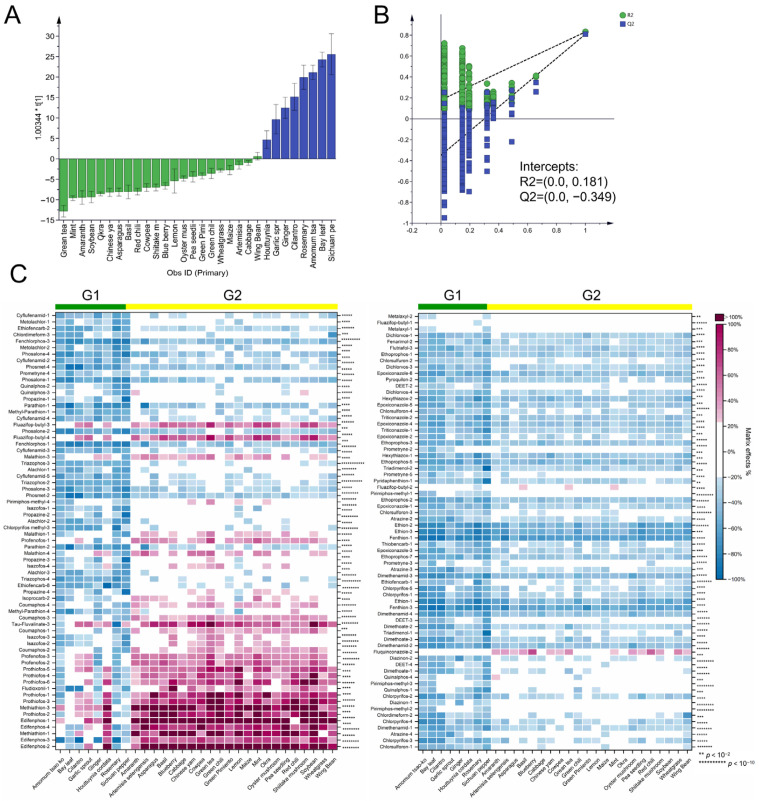
(**A**) The score plot of OPLS-DA modeling between groups G1 and G2; (**B**) the permutation test result (500 times); (**C**) the heatmap of MEs for 138 differential pesticide transitions between groups G1 and G2, 2 to 10 asterisks (** to **********) indicated significant difference (*p* < 0.01 to *p* < 10^−10^).

**Figure 4 foods-12-01226-f004:**
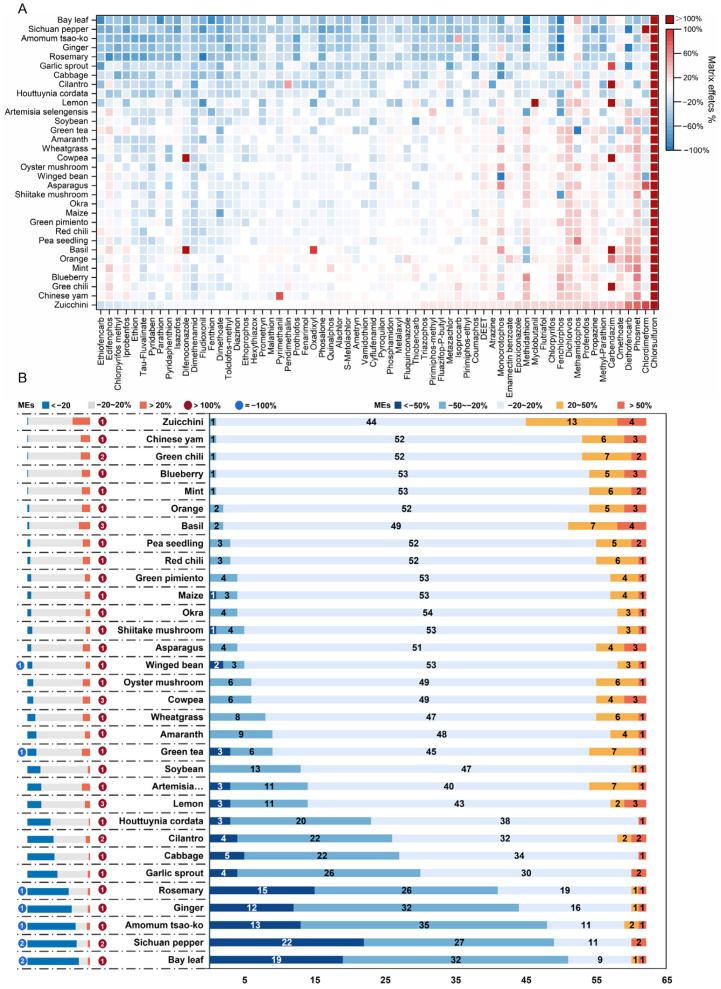
Distribution of matrix effects (MEs) on 62 pesticides in 32 matrices (**A**) and the comparison of MEs in different matrix species (**B**) in the TOF-MS scan under IDA mode.

**Figure 5 foods-12-01226-f005:**
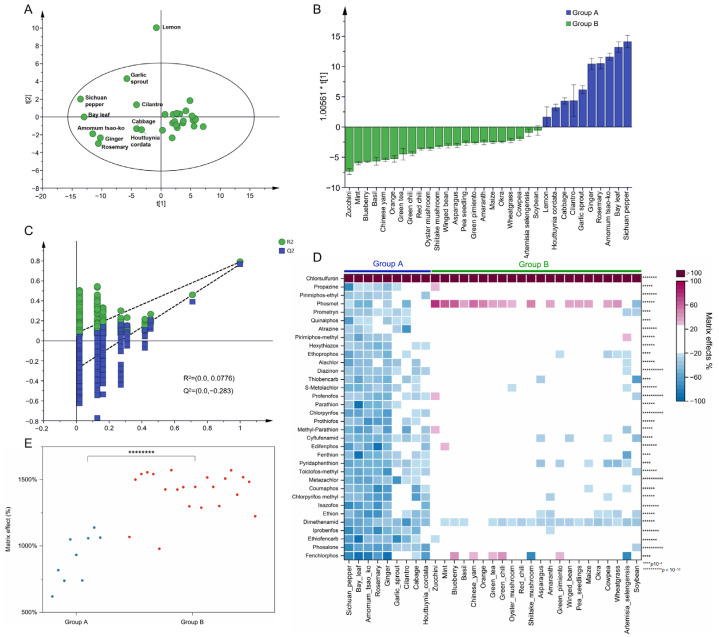
(**A**) The score plot of PCA modeling; (**B**) the score plot of OPLS-DA modeling between groups A and B; (**C**) the permutation test result (500 times); (**D**) the heatmap of MEs for 34 differential pesticides between groups A and B; (**E**) distribution of MEs on chlorsulfuron between groups A and B, and 4 to 10 asterisks (**** to **********) indicated significant difference (*p* < 10^−4^ to *p* < 10^−10^).

**Figure 6 foods-12-01226-f006:**
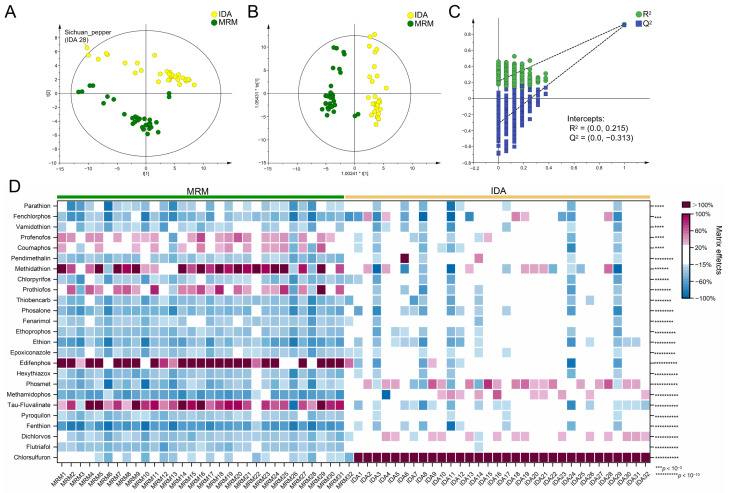
(**A**) The score plot of PCA modeling; (**B**) the score plot of OPLS-DA modeling; (**C**) the 500 times permutation test result; (**D**) the heatmap of MEs on 25 differential pesticides between the MRM scan and the TOF-MS survey scan under IDA mode, and 3 to 10 asterisks (*** to **********) indicated significant difference (*p* < 10^−3^ to *p* < 10^−10^).

**Figure 7 foods-12-01226-f007:**
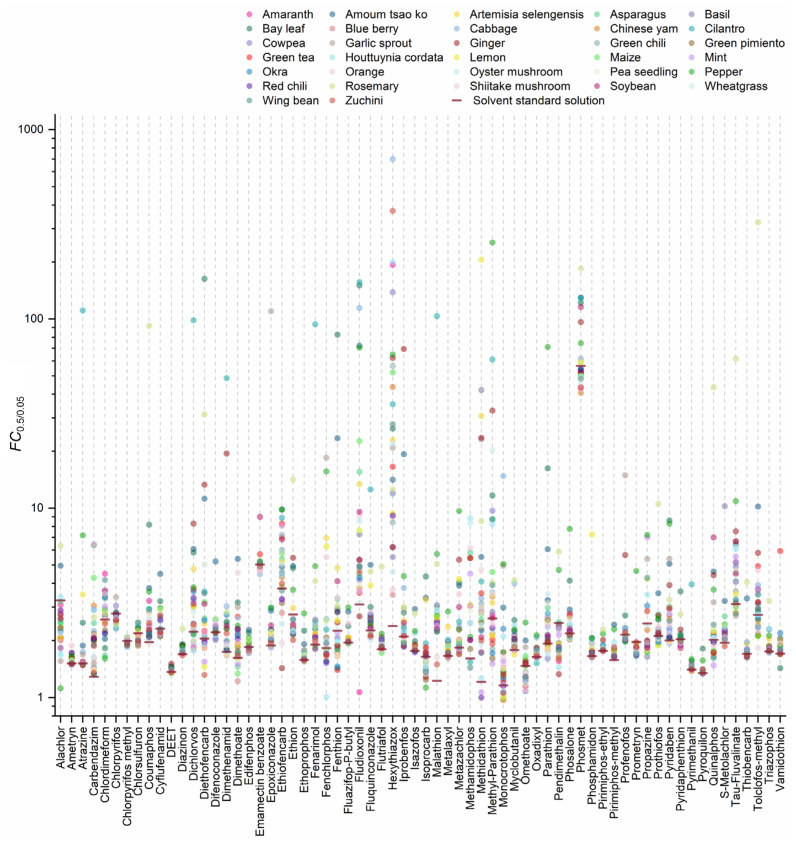
Distribution of FC_0.5/0.05_ for 62 pesticides among 32 matrix species.

## Data Availability

Data are contained within the article and Appendix A.

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
