# Peer review of "Impact of Matrix Species and Mass Spectrometry on Matrix Effects in Multi-Residue Pesticide Analysis Based on QuEChERS-LC-MS"

_foods, 2023, doi:10.3390/foods12061226_

Round 1
Reviewer 1 Report
Dear Authors,
Being invited to review this manuscript, I had a great pleasure to read and scrutinize one rather extraordinary, superb and original work from the field of mass spectrometry and matrix effects of different pesticides in various species.
The Authors show ME typing strategy drawing on the metabolomics analysis thinking and tools was achieved which providing an efficient strategy for analysts to handle rapid ME analysis with real-time combination of massive pesticide targets, different matrix species, and different LC-MS conditions. They analyzed pesticides between groups which were identified by OPLS-DA analysis and further validated by modified heat map analysis and t-tests. Bay leaf, ginger, rosemary, Amomum tsaoko, Sichuan pepper, cilantro, Houttuynia cordata and garlic sprout induced medium to strong suppression on 56% and 48% of pesticide targets under MRM and IDA scan, respectively. Switching mass spectrometry from MRM scan to IDA scan simultaneously weakened MEs on 24 pesticides to negligible levels in 32 different matrix species.
Overall, this manuscript is well organized and written while the very idea for research is great and has practical aspect. Interesting results were well presented. The length of the manuscript is appropriate. Discussion is detailed and gives answers to every aspect of experiments applied in this work. Overall impression is that this manuscript can be warmly recommended without additional review in Foods especially considering the scope and topics of this journal.
With kind regards!
Reviewer
Author Response
Dear reviewer,
Thank you very much for your time reviewing the manuscript and encouraging comments on the merits.
Thank you and best regards.
Yours sincerely,
Jie Chen
On behalf of all the authors
Reviewer 2 Report
Please provide the meaning of abbreviations at the first use, even if they are considered trivial or well-known.
The matrix effect observed in the case of specified matrix pesticide combinations provides useful guidance for practicing analysts.
The details of extraction and cleanup steps employed during sample processing are critical for the matrix effect and practical application of the results. In such cases, the procedures applied should be specifically referenced to the steps in FS1 in the article.
Please note that the link given for supplementary materials in the article was not active at the time of the review of the manuscript.
The extracted ion chromatograms are very illustrative. Therefore, they could be better emphasized I the main text.
Table S3: retention times indicated in the heading are not included in the table.
Figure S4: the legend is too small to be readable in the printed version.

Author Response
Dear reviewer,
Thank you for your letter and the comments concerning our manuscript. Those comments are all valuable and helpful for revising and improving our paper and the essential guiding significance of our research. We have studied the comments carefully and made the correction we hope to meet with approval. The revised portion is marked in red on the paper. The revisions and the responses to the reviewer’s comments are in the attachment.
Special thanks to you for your good comments.
We tried our best to improve the manuscript and made some changes in the manuscript. These changes will not influence the content and framework of the paper.
Once again, thank you very much for your comments and suggestions.
Thank you and best regards.
Yours sincerely,
Jie Chen
On behalf of all the authors

Reviewer 3 Report
The manuscript has partial merit. Please see the specific comments below. Only if these are addressed, it can be considered.
1) In my opinion the main bottleneck of the manuscript is its applicability. The authors should better describe how their approach will be of benefit, especially for routine laboratories. Otherwise, despite the work being conducted, it fails to convey its "message". For instance, how the IDA scan in HRMS will function "pragmatically" in parallel with MRM mode in targeted analysis. This is not clear. In addition, the differences obtained between the two modes, as presented, perplexes the situation instead of simplifying it.
2) Please cross-check matrix effects for specific substances (such as carbendazim) with the literature findings. Is there consistency with the obatined results? How many replicates of the samples were analyzed?
3) The authors report that the IDA scan reduced MEs for some compounds? But, what about the LOQs? This is very important, especially for the achievement of MRLs with HRMS instruments. Since the QTOF system used by the group is robust, I expect a comment on this aspect. Otherwise, the reduction of MEs is a reality, but its applicability is limited.
4) Did the authors test other modifications of QuEChERS? A comment is needed.
5) Please add the following reference in the discussion of your results
Makni et al., "Improving the monitoring of multi-class pesticides in baby foods using QuEChERS-UHPLC-Q-TOF with automated identification based on MS/MS similarity algorithms", in Food Chemistry, 395, 2022
6) Extensive editing is required. In its current form, the readability of the manuscript is cumbersome.
Other typos-rephrasing etc: Page 2, line 47, "Few studies have". Lines 51-53, need revision, and are difficult to follow.
Page 3, line 94, please rephrase formulations. You probably mean the composition of Bond Elut which can be provided with a reference. Same page, line 124, "analyte".
Use "species" throughout the manuscript not "specie".
Author Response
Dear reviewer,
Thank you for your comments and professional advice concerning our manuscript. Those comments are all valuable and helpful for revising and improving our manuscript and the critical guiding significance of our research. We have studied the comments carefully and made the correction we hope to meet with approval. Please see the the point-by-point response to the reviewer's comments in the attachment.
Special thanks to you for your good comments.
We tried our best to improve the manuscript and made some changes in the manuscript. These changes will not influence the content and framework of the paper.
Once again, thank you very much for your comments and suggestions.
Yours sincerely,
Jie Chen
On behalf of all the authors
